# Research on Expansion Characteristics of Aquaculture Ponds and Variations in Ecosystem Service Value from the Perspective of Protecting Cultivated Lands: A Case Study of Liyang City, China

**DOI:** 10.3390/ijerph19148774

**Published:** 2022-07-19

**Authors:** Bochuan Zhao, Yongfu Li, Yazhu Wang, Guoqing Zhi

**Affiliations:** 1Shanghai Academy of Fine Arts, Shanghai University, Shanghai 200444, China; 19722860@shu.edu.cn; 2Nanjing Institute of Geography and Limnology, Chinese Academy of Sciences, Nanjing 210000, China; 3College of Applied Arts and Sciences, Beijing Union University, No. 197 Beitucheng West Road, Beijing 100191, China; 191070510111@buu.edu.cn

**Keywords:** cultivated land, aquaculture pond, ecosystem service value (ESV), land use transformation

## Abstract

In the context of global food insecurity, a large amount of cultivated land in China has been occupied by aquaculture ponds, leading to a series of variations in the ecological environment. The Chinese government pays close attention to the problem. In order to achieve sustainable development and ensure the safety of China’s cultivated land, the paper uses Liyang City as an example to discuss the spatial characteristics of the expansion of aquaculture ponds through occupying cultivated lands and analyzes the variations in ecosystem service value and cultivated land function. The conclusions are as follows: (1) 2073.24 hectares of cultivated lands were occupied for expanding aquaculture ponds in Liyang from 2009 to 2019, and there was a small number of new aquaculture ponds in the ecological protection area, which shows that the aquaculture ponds in Liyang City are at the stage of disorderly expansion; (2) the total value of ecosystem services increased by 1.43%; supply and support services values decreased, but the increase in regulation and cultural services values was sufficient to more than compensate for the mentioned losses; and (3) the expansion of aquaculture ponds leads to a decrease in the carbon storage of cultivated land, which in turn has negative impacts such as an increase in atmospheric carbon concentration.

## 1. Introduction

In May 2021, the Food and Agriculture Organization (FAO) of the United Nations and the World Food Programme (WFP) issued the 2021 Global Report on Food Crisis. According to the report, the problem of food insecurity reached its maximum level in the previous five years and the increasing seriousness of the food crisis was influenced by COVID-19 and the international situation. Cultivated land is not only the main source of food production [1], but also an important guarantee for the steady development of the social economy [2,3]. Although cultivated land is a renewable resource [4], its recovery cycle is very long. Therefore, when a large amount of cultivated land is occupied by other land types, it will undoubtedly directly threaten national food security. With the development of urbanization and industrialization in China, cultivated lands experienced a rapid decline. Later, China successively issued the ecological civilization construction strategy and rural revitalization strategy. Under dual restraints of the ecological red line and the red line of cultivated land protection, the decline in the area of cultivated lands was reduced to a certain extent, but the amount of cultivated lands used for grain continued to decrease. The fact was that the phenomenon of non-grain production of cultivated land occurred with a structural adjustment in the agricultural industry [5,6]. The “non-grain production of cultivated land” refers to the behavior of planting non-food crops on cultivated lands, for example, planting economic crops, excavating aquaculture ponds and afforestation. All these behaviors will affect the quality of cultivated land and may threaten the security of cultivated land and food in cases of disordered development. The behavior of excavating aquaculture ponds has been widely considered as an important cause of the decrease in cultivated lands [7]. Increasingly more cultivated lands are occupied by aquaculture ponds [8]. It is estimated that globally, about 1.5 million hectares of land is occupied by aquaculture every year [9]. The rapid development of aquaculture ponds depends on occupying large areas of high-quality cultivated land. Although it has created huge economic benefits, it also has some negative influences on the ecological environment [10,11,12,13,14]. In this context, the paper quantitatively evaluates the spatial distribution and variations in the characteristics of aquaculture ponds, which is important and significant for the protection of regional cultivated lands and the sustainable development of agriculture.

As one of the main utilization types of terrestrial ecosystems, cultivated lands play an important role in ensuring human and food security, and variations in cultivated land have an important influence on the regional ecological environment [15]. After issuing the reform scheme of rural land circulation in 2008, China saw rapid development in rural land circulation, the utilization type of cultivated lands also emerged [16], and transformation from cultivated lands into aquaculture ponds gradually became a prominent utilization mode in the transformation of cultivated lands. Some studies indicate that the expansion of aquaculture ponds in the Yellow River Delta was achieved through occupying cultivated lands [17]. Some other scholars found that a notably large amount of cultivated land was transformed into aquaculture ponds through their analysis of the spatial variation in aquaculture ponds in the coastal area of China [18]. Due to its speed and low cost [19], remote sensing data is widely used in studying aquaculture ponds. Some scholars supervised and classified the land cover of aquaculture ponds through remote sensing data [20] and some other scholars plotted the distribution of aquaculture ponds in the coastal area of the East China Sea through TM imagery [21]. In addition, different scholars studied aquaculture ponds from different perspectives, for example, hazards of aquaculture ponds in Southeast Asia in closed water areas [22]; expansion of aquaculture ponds along the coastline of China [23]; spatial distribution of aquaculture ponds along the coastline in India [24]; the influence of aquaculture on the ecological environment along the coastline in Bangladesh [25]; etc. In terms of research methods, many scholars tend to recognize the spatial variation in aquaculture ponds based on the remote sensing satellite data [26] or automatically extract positions of aquaculture ponds from Google image data [27]. Some scholars studied the driving factors of occupying cultivated lands for the expansion of aquaculture ponds in Guangdong Province through questionnaire data [28]. Some scholars analyzed land use variations in the coastal area due to aquaculture ponds through remote sensing (RS) and geographic information system (GIS) [29]. Some scholars analyzed the degree of influence of the expansion of aquaculture ponds on changes in local land cover through the landscape pattern index [9].

Another emphasis in aquaculture pond research is problems in the ecological environment due to the expansion of aquaculture ponds. Many studies indicate that the expansion of aquaculture ponds can promote local economic development but, to a certain extent, will also have a negative influence on the local ecological environment [30,31]. The long-term negligence of ecological value and social value of cultivated land resources is the largest shortcoming and flaw in protecting cultivated lands in China [32]. The rapid development of aquaculture ponds through occupying cultivated lands leads to further degradation of the ecological environment. Some scholars studied the problem from the view of ecosystem service. The ecosystem service concept first came into view in the 1980s [33]. The ecosystem service refers to life support products and services that are obtained directly or indirectly through the structure, process and function of the ecosystem [34]. The ecosystem service is the material basis and basic condition for the survival and development of humans and is the key natural capital of humans [35]. The ecosystem can provide necessary food and raw materials for humans and can support non-material services such as biological and chemical circulation and carbon sequestration [36]. The decisions of policy-makers are often influenced by economic data. Since ecosystem services cannot be directly measured by economic data, they must be quantified in economic terms based on their price in the market or the price of alternative goods and services [37]. Ecosystem service value (ESV) is an important quantitative and evaluation method for the function of ecosystem services [38,39]. The change in ESV is related to factors such as land use change, climate change and biodiversity [40], among which land use change is the most direct reflection of human activities. Therefore, ESV measurement methods are widely used in related research on land use change, such as the impact of land use change on the ecological environment in the process of urbanization [41,42]. In addition, ESV has good practical application value in the selection of regional sustainable development strategies [43], urban planning and management [44] and the formulation of ecological compensation policies [45].

In the context of paying close attention to the ecological environment and protection of cultivated lands in China, evaluating the influence of transformation between aquaculture ponds and cultivated lands on ESV is important and significant for the resource protection and sustainable development of cultivated lands. The paper aims to understand the spatial characteristics of the expansion of aquaculture ponds through occupying cultivated lands in the countryside of China and to analyze the influence of the process on local ESV and the function of cultivated lands. The paper takes Liyang City (Jiangsu Province) as an example to investigate the objectives of: (1) analyze the spatial characteristics and quantitative relationship of the expansion of local aquaculture ponds through occupying cultivated lands from 2009 to 2019; (2) analyze local ESV change from the perspective of service function and land use type, and analyze the value flow of the ecosystem service due to the expansion of aquaculture ponds through occupying cultivated lands; and (3) use the carbon capacity to evaluate the influence of the expansion of aquaculture ponds through occupying cultivated lands on the function of cultivated lands to provide a reference for promoting the sustainable utilization of regional resources of cultivated lands.

## 2. Materials and Methods

### 2.1. Overview of Research Area

Located in eastern China, Liyang City is a county-level city in the southwest of Jiangsu Province, as shown in Figure 1. With the northern latitude 31°09′~31°41′ and eastern longitude 119°08′~119°36′, Liyang has a length of 59.06 km from north to south, width of 45.14 km from east to west and a total land area of 1535 km^2^. Located at the edge of the northern subtropical region, Liyang has a maritime, moist and monsoon climate and significant monsoon characteristics. The prevailing wind direction throughout the year is the easterly wind, annual average temperature is 17.5 °C and annual average precipitation is 1149.7 mm. Liyang City is located in complex topography of mountainous land, hills, plains and polder areas. It has plains from west to east with flat terrain, and mountainous lands and hills in the west, north and south.

Liyang City is a nationally advanced county in food production. However, many cultivated lands are uncultivated or occupied by aquaculture ponds, and large areas of non-food crops have been cultivated in recent years (Figure 2a). The per capita area of cultivated land decreased from 1.05 mu to 0.96 mu, which is far lower than the national per capita level of 1.51 mu. Although Liyang has gained the honorable title of “National Ecological City”, it still faces many challenges in the ecological environment; the problem of organic matter and soil pollution is gradually increasing; the situation of environmental water protection is serious (Figure 2b) and direct drainage of agricultural wastewater is still prominent; and the ecological environment indicates a significant trend of environmental degradation, and fragmentation of ecological space tends to be serious.

### 2.2. Data Source and Processing

The 2009 land use data for Liyang City were taken from The Second China’s Territory Survey Data, while the 2019 land use data and boundary data of administrative districts were taken from The Third China’s Territory Survey Data. The vector data of the ecological protection area were provided by the Liyang Ecological Environment Bureau. The cultivated area of food crops, yield and dosage of chemical fertilizer and pesticide were taken from the Liyang Statistical Yearbook. The national average price of food crops was taken from the Collection of National Cost Benefits of Agricultural Products in 2019.

### 2.3. Research Methods

#### 2.3.1. ESV Evaluation

The proposed value coefficient by Costanza et al. [38] has been widely used in research on ESV, but the value coefficient is not universally agreed. To better analyze the value variation in ecosystem services in the research area, the study assigns value equivalence for ecosystem services on cultivated lands and aquaculture ponds in the research area based on existing research findings [46] and the amended “Table of value equivalence for ecosystem service per unit area for China’s land ecosystem” by Xie Gaodi et al. [47] based on actual conditions in China (Table 1); and calculates ESV in the research area through Formulas (1) and (2):(1)$$\mathrm{ESV}=\overset{n}{\underset{i=1}{\sum }}A_{i}\times VC_{i}$$
(2)$$VC_{i}=\overset{k}{\underset{j=1}{\sum }}EC_{j}\times E_{a}$$
where, ESV is the ecosystem service value, Yuan/a; *i* is the land use type; *j* is the service type of the ecosystem; $A_{i}$ is the area of type *i* land use, ha; *VC_i_* is the ESV per unit area of type *i* land use, Yuan/hm^−2^ a^−1^; *EC_j_* is the value equivalence of ecosystem service for type *j* land use; *Ea* is the economic value of ecosystem service per unit, Yuan/hm^−2^ a^−1^.

To accurately determine the economic value of ecosystem services per unit in the research area, the study corrects related parameters through Formula (3) [48]. The obtained the economic value of natural food production on farmland in the research area in 2019, which is the economic value of ecosystem serviced per unit, is RMB 4502.99 Yuan/hm^−2^ a^−1^.
(3)$$E_{a}=1/7\overset{n}{\underset{i=1}{\sum }}\frac{m_{i}q_{i}p_{i}}{M}\times MCI$$
where, *Ea* is the economic value of ecosystem services per unit in the research area, Yuan/hm^−2^a^−1^; *i* is the type of food crop; $m_{i}$ is the national average price of type *i* food crop, Yuan/kg; $q_{i}$ is the cultivated area of type *i* food crop, hm^2^; $p_{i}$ is the per unit yield of type *i* food crop, kg/hm^2^; *M* is the total cultivated area of food crop, hm^2^; *MCI* is the multi-cropping index of farmland in the research area, set at 2.5.

#### 2.3.2. Value Flow Analysis for Ecosystem Services

The value flow analysis of ecosystem services calculates profit and loss in ESV due to mutual transformation between different land use types through transformation data of land use types. The purpose is to analyze the influence of land use transformation on ESV [49]. The calculation is shown in Formula (4):(4)$$PL_{ij}=\left(VC_{j}-VC_{i}\right)\times A_{ij}$$
where, $PL_{ij}$ is the profit and loss in ESV after transforming type *i* land use into type *j* land use; $VC_{j}$ and $VC_{i}$ are value coefficients of ecosystem services for type *i* land use and type *j* land use; $A_{ij}$ is the area of transforming type *i* land use into type *j* land use.

#### 2.3.3. Evaluation Method for Functions of Cultivated Lands

The expansion of aquaculture ponds through occupying cultivated lands and drainage of aquaculture wastewater will, to a certain extent, have a negative influence on the quality of cultivated lands. Due to a lack of soil data, quantitative evaluation cannot be directly made for soil texture. Hence, the study uses the carbon capacity of cultivated lands to evaluate the functional variation in cultivated lands.

The soil organic carbon is an important index for representing the variation in soil fertility, which influences the physical, chemical and biological characteristics of soil and lays an important foundation for the stable production of food crops and sustainable development of agriculture. The carbon capacity of different types of land use is represented by the product between carbon density of soil and the area of related land use type, as shown in Formula (5):(5)$$CS_{i}=CD_{i}\times A_{i}$$
where, $CS_{i}$ is the carbon capacity of type i land use, kg; $CD_{i}$ is the carbon density of type i land use, kg/m^2^; $A_{i}$ is the area of type  i land use, m^2^. In Jiangsu Province, the carbon density (0~100 cm) is 9.29 kg/m^2^ for cultivated lands and 8.11 kg/m^2^ for aquaculture ponds [50,51].

## 3. Results

### 3.1. Characteristic Analysis for Cultivated Lands and Aquaculture Ponds

#### 3.1.1. Quantitative Variation in Aquaculture Ponds

The agricultural area in Liyang City was 59,797.42 hectares in 2009 but decreased by 14,887.47 hectares to 44,909.95 hectares in 2019, where 6279.21 hectares of cultivated lands was occupied by aquaculture lands (new) and 4205.97 hectares of aquaculture ponds was transformed into cultivated lands (decrease in aquaculture ponds). In fact, 2073.24 hectares of cultivated lands was occupied for expanding aquaculture ponds from 2009 to 2019.

The increased and decreased areas of aquaculture ponds in various towns are different. See Table 2 for detailed changes. The area of aquaculture ponds increased by 3675.99 hectares in Licheng Town, Shanghuang Town, Tianmu Lake Town, Shangxing Town and Shezhu Town; but decreased by 1602.75 hectares in Daitou Town, Bieqiao Town, Zhuze Town, Nandu Town and Kunlun Street. The greatest increase in aquaculture pond area occurred in Shezhu Town with about 2478.99 hectares, which was 39.48% of the total increased areas of aquaculture ponds in the city. The smallest decreased in aquaculture pond area was in Bieqiao Town with about 890.00 hectares, being 21.16% of the total decreased areas of aquaculture ponds in the city.

#### 3.1.2. Spatial Expansion of Aquaculture Ponds

As shown in Figure 3, new aquaculture ponds in Liyang City are mainly distributed in Shezhu Town and Shangxing Town in the southwest, as well as Shanghuang Town in the northeast corner of Liyang City. Although significant and concentrated distribution is not found in the other towns, some small and fragmented spots can be seen in each town. The decreased aquaculture ponds are mainly distributed in the north of Liyang City, including Daitou Town, Bieqiao Town, Zhuze Town and Nandu Town, and some aquaculture ponds can also be found in the southern Daibu Town and Shezhu Town.

Generally, the ecological protection area is the area that requires mandatory protection due to ecological sensitivity [52]. According to laws and regulations in China, development and productive construction activities are not allowed in ecological protection areas. As shown in Figure 3, however, some new aquaculture ponds are located in ecological protection areas. After calculation, there are 540.28 hectares of new aquaculture ponds in ecological protection areas, indicating a disordered expansion of aquaculture ponds in Liyang City.

### 3.2. ESV Change Analysis

#### 3.2.1. ESV Change

The expansion of aquaculture ponds through occupying cultivated lands leads to ESV change in Liyang City. From 2009 to 2019, ESV increased from RMB 13.821 billion Yuan to RMB 14.019 billion Yuan and total EVS increased by RMB 198 million Yuan.

The dominant type of ecosystem service function is regulation services. From 2009 to 2019, the regulation service value was over 90% of the total ecosystem service value and increased by RMB 293 million Yuan; the cultural service value increased by RMB 4.19 million Yuan; the supply service value and support service value decreased by RMB 92 million Yuan and RMB 8 million Yuan, respectively. See Table 3 for detailed changes.

From the view of the ecological value of land use type (Table 4), the ESV of cultivated lands is higher than aquaculture ponds. In the research period, the ecological service value decreased by RMB 437 million Yuan for cultivated lands and increased by RMB 691 million Yuan for aquaculture ponds; but the regulation service value decreased the most for cultivated lands, up to RMB 307 million Yuan, followed by the supply service value, about RMB 87 million Yuan; the decreased cultural service value was lowest, about RMB 6 million Yuan. For aquaculture ponds, the regulation service value increased the most, up to RMB 653 million Yuan, but the supply service value increased the least, only RMB 8 million Yuan. Hence, the expansion of aquaculture ponds through occupying cultivated lands can lead to a decrease in supply service value and support service value but an increase in regulation service value and cultural service value in the ecosystem in Liyang City.

#### 3.2.2. Value Flow of Ecosystem Service

From 2009 to 2019, due to the occupation of cultivated lands by aquaculture ponds, the ESV increased by RMB 1.03 billion Yuan in Liyang City, where the ESV increased by RMB 3.13 billion Yuan for the transformation of cultivated lands into aquaculture ponds and decreased by RMB 2.1 billion Yuan for the transformation of aquaculture ponds into cultivated lands. The main function leading to ESV change is the regulation function, of which there is a 95.92% change in total ESV in Liyang City.

From the view of ESV, expansion of aquaculture lands through occupying cultivated lands can improve the total ESV of the region. From the view of protection of cultivated lands and food security, the expansion of aquaculture lands through occupying cultivated lands will firstly lead to a quantitative decrease in cultivated lands and then lead to a quality decrease in cultivated lands. In the context of global food insecurity, the problem of protecting cultivated lands cannot be neglected.

### 3.3. Analysis of Functional Change in Cultivated Lands

Carbon emissions come mainly from human activity, including combustion of fossil fuels and land use change [53]. Different types of land use have different carbon storage capacities. The transformation of land use from a type with high carbon storage capacity into a land use type with low carbon storage capacity generally leads to carbon emissions into the atmosphere [54], which will threaten the regional ecosystem.

In Liyang City, the expansion of aquaculture ponds leads to the rapid discharge of organic carbon. The transformation from cultivated lands into aquaculture ponds leads to a decrease in soil carbon capacity of 8.74 × 10^7^ kg. The transformation from aquaculture ponds into cultivated lands leads to an increase in soil carbon capacity of 5.86 × 10^7^ kg. The expansion of aquaculture ponds leads to a decrease in soil carbon capacity of 2.89 × 10^7^ kg. The reduced emission volume of stored carbon storage into the atmosphere leads to further improvement in carbon concentration in the atmosphere.

## 4. Discussion

### 4.1. Causes of Expansion for Aquaculture Ponds

In recent years, the Chinese government has gradually paid close attention to the problem of protecting cultivated lands. In the early stage, many cultivated lands were occupied for construction due to the rapid development of industrialization and urbanization in China. With social and economic development, the disordered expansion of construction lands is effectively controlled and replaced by the expansion of aquaculture ponds. From 2009 to 2019, aquaculture lands rapidly expanded by occupying cultivated lands, and in Jiangsu Province there was a significant increase in the area of aquaculture ponds in Liyang City [18]. Aquaculture ponds have been found to be one of the land use types with rapid growth. Similarly, some other studies also indicate that the main land use type for promoting an increase in aquaculture ponds is cultivated land [17]. Although the expansion of aquaculture ponds can create enormous economic benefits, it also leads to a number of environmental problems [55].

In fact, the earnings of transforming cultivated lands into aquaculture ponds are far higher than grain earnings due to the low earnings of food crops on cultivated lands. According to an interview with local aquaculture households in Liyang City, the earnings per hectare of cultivated land is less than RMB 2895.97~3787.03 Yuan, while the earnings per hectare of aquaculture can reach RMB 119,402.99~149,253.73 Yuan. For this reason, the peasants are not willing to plant food crops and choose to transform cultivated lands into aquaculture ponds. Due to the influence of surrounding peasants [56], increasingly more peasants prefer to transform cultivated lands into aquaculture lands, which is the main reason for the rapid expansion of aquaculture ponds in Liyang City. Because there is unlimited control over cultivated lands in the Land Management Law of the People’s Republic of China, the first-level managers cannot effectively control the transformation from cultivated lands to aquaculture ponds by way of supervision and law enforcement, which is another important reason for the expansion of aquaculture ponds.

### 4.2. Influence of Expansion of Aquaculture Ponds

At present, two methods can be used to study ESV, including the method based on the equivalence factor of unit area value and that based on the price of unit service functions [48]. The method based on the price of unit service functions requires more accurate ecological data, while the method based on the equivalence factor of unit area value requires only area data of different land use types. In view of data availability, the equivalence factor method is widely used in China. The ESV difference has a strong correlation with the list of used equivalence factors. To obtain more accurate results, the paper corrects the ESV equivalence factor based on the actual conditions of Liyang City. LEI Jun-zheng et al. used the list of equivalence factors of XIE Gaodi and obtained consistent results with this study [49].

In 2009, the aquaculture of freshwater shrimps greatly promoted local economic development and solved employment, to a certain extent, in Shezhu Town. Later, Liyang City considered aquaculture as the dominant type of agricultural industry for eliminating poverty. From 2009 to 2019, the dominant land use transformation in Liyang City was from cultivated lands into aquaculture ponds. Related studies indicate that aquaculture ponds include large areas and rapid growth [57]. The expansion of aquaculture ponds leads to an increase in total ESVs in Liyang City due to a lower ESV per unit area of the cultivated area than aquaculture ponds. Some other studies indicate that an expansion in water domain land will lead to an increase in regional ESV [58]. Although the expansion of aquaculture ponds through occupying cultivated lands can improve regional ESV, it will also have some negative influences, which is neglected in previous studies. Most of the previous studies consider the ecosystem of a region as a whole and study the ecosystem of different land use types as an integral part but neglect differences among different land use types in the ecosystem. Some studies found that cultivated lands would be occupied for the expansion of aquaculture ponds [59], but few studies estimated the ESV change. In consideration of the urgency of the phenomenon, the study selects cultivated lands and aquaculture ponds to independently analyze their ESV changes, which will improve research in the field.

The example of expanding aquaculture ponds through occupying cultivated lands is not specific to Liyang City. Some studies indicate that aquaculture ponds are also expanded through occupying cultivated lands in the Yangtze River Delta of China [17], Sinaloa, Mexico [9] and Thailand [60]. Farmers need to regularly change the water in ponds in order to improve the breeding output. However, the breeding wastewater will often be discharged onto cultivated land near the pond, and the sediment in the wastewater will lead to the degradation of cultivated land, which is ultimately not suitable for grain planting [61]. The transformation from cultivated lands into aquaculture ponds will promote regional carbon emission, which conflicts with existing low-carbon policy and will not promote the long-term sustainable development of the local ecosystem. During the research period, although some aquaculture ponds were transformed into cultivated lands in Liyang City, newly cultivated lands had low grain production capacity [62] and could not maintain their original grain production capacity.

When studying the problem of expanding aquaculture ponds through occupying cultivated lands, we shall consider that it can improve regional ESV and threaten the production capacity of cultivated lands. On the practical level, regional managers shall comprehensively consider the ESV and the protection of cultivated lands based on regional development objectives.

### 4.3. Limitations and Innovations

The different classification of land use types may lead to different ESV results. Although Costanza [38] and XIE Gao-di [63] provided value equivalence of different land use types, their classifications did not include aquaculture ponds. Hence, the paper uses ESV equivalence of the water area only to substitute aquaculture ponds, but the water domain land includes multiple Grade 2 classifications such as lakes, aquaculture ponds, reservoirs and rivers, which may result in certain deviations from the actual conditions in Liyang City. Due to the lack of soil texture data, the change in cultivated lands can be analyzed only through carbon capacity. Despite the above problem, it is possible to determine the influence of the expansion of aquaculture ponds on ESV at the macroscopic level. Therefore, the paper aims to attract the attention of scholars to a series of problems due to the expansion of aquaculture ponds through occupying cultivated lands.

Most previous studies analyze the spatial distribution of aquaculture ponds along the coastline of a region based on macro-scale or medium-scale remote sensing image data, instead of studying aquaculture ponds and cultivated lands as a whole. In addition, ESV-related studies usually analyze all land use types as a whole, which may overlook regional characteristics. For the above problem, the paper analyzes the expansion of aquaculture ponds through occupying cultivated lands and its ESV change in Liyang City through vector data based on the perspective of a microcosmic city. The paper firstly refined previous studies in the research scale, then used the vector data with higher accuracy than raster data, and finally considered aquaculture ponds and cultivated lands as a whole to offset previous deficiencies.

## 5. Conclusions

Based on the land use data of Liyang City in 2009 and 2019, the paper analyzes variation in the ESV and function of cultivated lands due to occupation of cultivated lands by aquaculture ponds for expansion and draws conclusions as below:

Large areas of cultivated land is occupied for expanding aquaculture ponds in Liyang City. The increased aquaculture ponds are mainly distributed in the southwest and northeast corners, while decreased aquaculture ponds are mainly distributed in the north of Liyang City. It is noted that new aquaculture ponds in the ecological protection area indicate disordered expansion of aquaculture ponds.

The expansion of aquaculture ponds through occupying cultivated lands will lead to an increase in total ESV in Liyang City, where values of supply services and support services will decrease but values of regulation services and cultural services will increase. The regulatory function is the main factor that results in the value variation in ecosystem services.

Although the expansion of aquaculture ponds through occupying cultivated lands can improve Liyang City’s ESV, it will have a negative influence on cultivated lands. During the research period, the carbon storage of cultivated land in Liyang City significantly decreased, resulting in a large volume of carbon emissions into the atmosphere, which will have an adverse influence on the atmospheric environment and sustainable development of agriculture.

## Figures and Tables

**Figure 1 ijerph-19-08774-f001:**
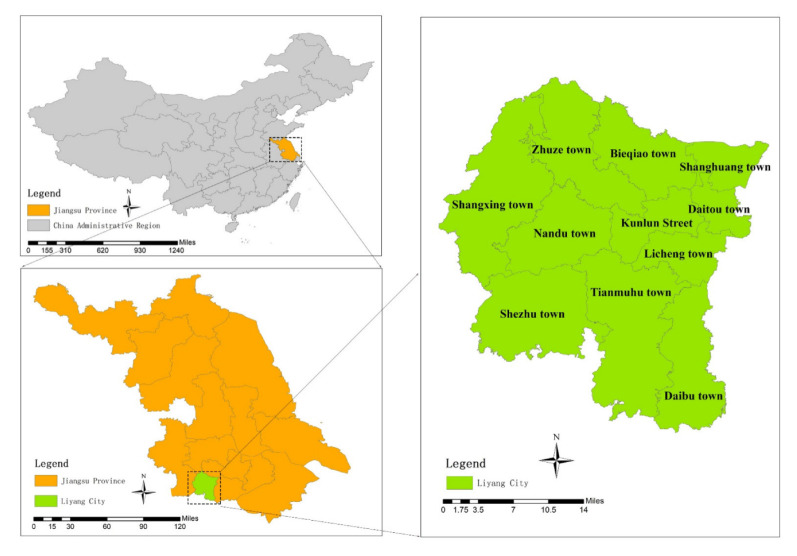
Geographical position of Liyang City.

**Figure 2 ijerph-19-08774-f002:**
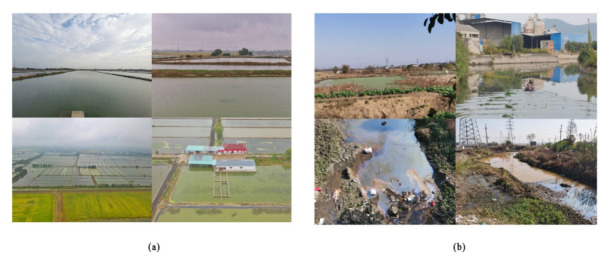
Status quo of aquaculture ponds and ecological environment. (**a**) Aquaculture ponds and (**b**) contaminated water.

**Figure 3 ijerph-19-08774-f003:**
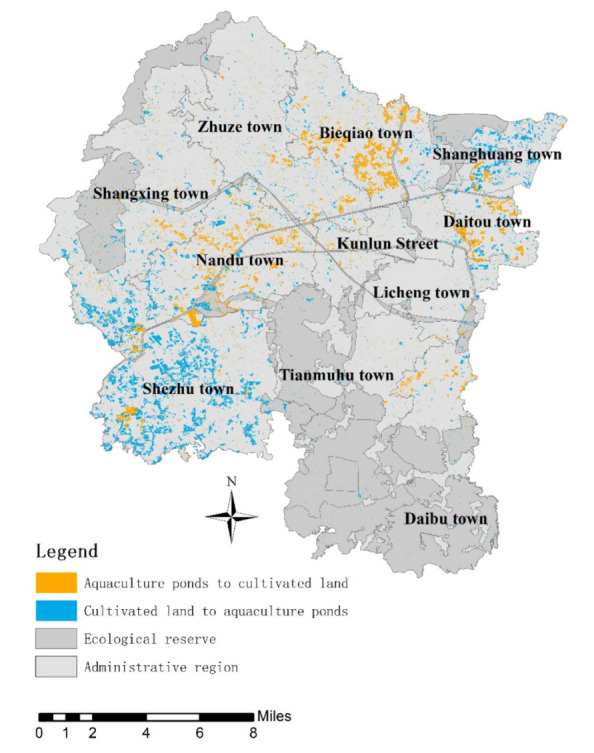
Spatial expansion and distribution of aquaculture ponds in Liyang City.

**Table 1 ijerph-19-08774-t001:** Value equivalence of ecosystem services.

Grade 1 Ecosystem Service	Grade 2 Ecosystem Service	Cultivated Lands	Aquaculture Ponds
Supply service	Food production	1.36	0.8
	Raw materials production	0.09	0.23
Regulation service	Gas regulation	1.11	0.77
	Climate regulation	0.57	2.29
	Environmental purification	0.17	5.55
	Hydrological regulation	2.72	102.24
Support service	Soil conservation	0.01	0.93
	Maintain nutrient cycling	0.19	0.07
	Biological diversity	0.21	2.55
Cultural service	Aesthetic landscape	0.09	1.89

**Table 2 ijerph-19-08774-t002:** Change in aquaculture ponds in each town of Liyang City from 2009 to 2019.

Towns	Transformation from Aquaculture Ponds into Cultivated Lands/ha (Area Decrease of Ponds)	Transformation from Cultivated Lands into Aquaculture Lands/ha (Area Increase of Ponds)	Actual Change in Aquaculture Ponds/ha
Licheng Town	71.75	124.32	52.56
Shanghuang Town	114.52	617.89	503.37
Tianmu Lake Town	61.01	136.30	75.29
Shangxing Town	363.60	929.37	565.78
Shezhu Town	361.84	2840.83	2478.99
Daitou Town	378.53	186.40	−192.13
Daibu Town	173.36	104.37	−68.99
Bieqiao Town	1235.68	345.68	−890.00
Zhuze Town	325.55	267.95	−57.60
Nandu Town	995.29	621.41	−373.88
Kunlun Street	124.83	104.68	−20.15
Total	4205.97	6279.21	2073.24

Note: In China’s administrative divisions, the level of streets and towns is the same.

**Table 3 ijerph-19-08774-t003:** ESV of different service functions in Liyang City.

Classification of Ecosystem Service	2009		2019		ESV Change/RMB 100 Million Yuan
	ESV/RMB 100 Million Yuan	Proportion	ESV/RMB 100 Million Yuan	Proportion	
Supply service	4.96	3.59%	4.05	2.89%	−0.92
Regulation service	126	91.39%	129	92.19%	2.93
Support service	4.75	3.44%	4.67	3.33%	−0.083
Cultural service	2.19	1.58%	2.23	1.59%	0.042
Total	138.21	100.00%	140.19	100.00%	1.98

**Table 4 ijerph-19-08774-t004:** ESV of cultivated lands and aquaculture ponds in Liyang City.

**Grade 1**	Grade 2	2009	2019	2009	2019
		ESV of Cultivated Lands/RMB 100 Million	ESV of Cultivated Lands/RMB 100 Million	ESV of Aquaculture Ponds/RMB 100 Million	ESV of Aquaculture Ponds/RMB 100 Million
Supply service	Food production	3.66	2.75	0.82	0.89
	Raw materials production	0.24	0.18	0.24	0.25
Regulation service	Gas regulation	2.99	2.24	0.79	0.83
	Climate regulation	1.53	1.15	2.36	2.48
	Environmental cleaning	0.46	0.34	5.71	6.01
	Hydrological regulation	7.32	5.5	105	111.07
Support service	Soil conservation	0.03	0.02	0.96	1.01
	Maintenance of nutrient circulation	0.51	0.38	0.07	0.08
	Biological diversity	0.56	0.43	2.62	2.76
Cultural service	Aesthetic landscape	0.24	0.18	1.94	2.05
Total		17.54	13.17	120.51	127.42

## Data Availability

Social and economic data are from the official website of Liyang municipal government.

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
