# Peer review of "Research on Expansion Characteristics of Aquaculture Ponds and Variations in Ecosystem Service Value from the Perspective of Protecting Cultivated Lands: A Case Study of Liyang City, China"

_ijerph, 2022, doi:10.3390/ijerph19148774_

Round 1

Reviewer 1 Report

GENERAL COMMENTS

I read the manuscript entitled "Research on Expansion Characteristics of Aquaculture Ponds 2 and Variation of Ecosystem Service Value from the Perspective 3 of Protecting Cultivated Lands: A Case Study of Liyang City, 4 China". The text is understandable and includes the necessary information. Nevertheless, there are several grammatical errors (see specific comments), some concepts need to be clarified (see specific comments), and I had trouble following a few sentences (see specific comments). Therefore, I think detailed copyediting would be helpful.

The comprehensive literature review shows that the addressed theme is challenging and topical. The used methods are suited, though they can be improved, and the study is somewhat innovative in what concerns the research scale. The results are presented clearly and straightforwardly, though the conclusion that the expansion of aquaculture ponds leads to a decrease in the production capacity of the cultivated land per unit of area is very questionable.

Although providing some advances in this field of research, this work's results are not ground-breaking and have some methods limitations, as recognised by the authors in the discussion section, and a significant conclusion is very debatable. Nevertheless, this is honest and valid research that contributes valuable information. Therefore, all considered, I cannot advise against the manuscript publication. However, if the authors are willing to make Major Revisions, I recommend considering the following specific comments.

SPECIFIC COMMENTS

1-      Detailed copyediting would be helpful to avoid grammatical errors and make the text more precise and easier to understand. Here is an example concerning the second conclusion stated in the Abstract (lines 23-25):

Instead of

"(2) Ecosystem service value in the period increased by 1.23%, of which the value of supply services and support services decreased, while the value of regulation service and cultural service.” (This sentence in incomplete and makes no sense).

consider

" (2) The total value of ecosystem services increased by 1.23%; supply and support services values decreased, but the increase in regulation and cultural services values was enough to more than compensate for the mentioned losses."

2-      It is not explained how the increase of 1.23% mentioned in the previous sentence was determined. However, according to the values in table 4 (138,21 in 2009 and 140,19 in 2019), this increase would be 1.43%.

3-      In lines 36-38, the authors say that the “cultivated lands” … “ensure ecological security as non-renewably precious resources”. However, according to conventional natural resources classification [see for example Owen (1980)] soil fertility and agricultural products, including vegetables, grains, fruits, and fibres, are considered renewable resources. It is essential to clarify the concept of renewable natural resources, which may be exhaustible if not correctly used, because it has direct implications for some of the draw conclusions.

4-       In line 164, instead of “… I is the land use type;” consider “i is the land use type;”

5-      In line 173, instead of “… where EA is the economic value …” consider “… where Ea is the economic value”.

6-      In line 212, instead of “The agricultural acreage was 5979,42 hectares …” consider “The agricultural area was 5979, 42 ha …”. Acreage is the extended area in acres, not in hectares (symbol: ha).

7-      In Table 3, in the last row of the Towns column appears “Kunlun Street”. Can Kunlun Street be considered a town?

8-      The claim made in the following sentence (lines 291-297) is not supported by the results and is strongly speculative: “The per unit area grain yield, and per unit area dosage of chemical fertilizers and pesticides for cultivated lands are calculated, as shown in Table 6. In the period of research, the per unit area dosage of chemical fertilizers and pesticides increases slightly, but per unit area grain field decreases for cultivated lands. It indicates that the discharge amount of aquaculture wastewater increases with the expansion of aquaculture ponds to further have a negative influence on the quality of cultivated lands. Specifically, the per unit area grain yield of cultivated lands decreases.”  In fact, during the last decades, 30% of agricultural land (1.5 billion hectares) has been abandoned due to erosion and degradation. Therefore, agricultural soil, though renewable is properly managed, is easily exhaustible. It degrades at a much faster rate than its natural regeneration rate, which is a much slower process; approximately 500 years are required to “rebuild” 25 mm of lost soil by erosion. Of all the factors, the one that most contributes to soil loss through erosion and its degradation is its intense and continuous use. “Data drawn from a global compilation of studies quantitatively confirm the long-articulated contention that erosion rates from conventionally ploughed agricultural fields average 1–2 orders of magnitude greater than rates of soil production, erosion under native vegetation, and long-term geological erosion.” (Montgomery, 2007).

9-       Pay attention to the “Error” mentioned in line 370.

1-  The sentence in lines 371-372, “The soil will be degraded due to the expansion of aquaculture ponds and finally not applicable to crop planting [57]”, is not supported by the results (see comment 9).  Does the paper concerning citation 57 support this claim regarding the study area?

Montgomery, D. R. (2007). Soil erosion and agricultural sustainability. Proceedings of the National Academy of Sciences104(33), 13268-13272.

Owen, O.S. (1980). Natural Resources Conservation: An Ecological Approach (3 rd ed.. New York: Macmillian

Reviewer 2 Report

Dear Authors,

The aim of the paper was to understand the spatial characteristics of the expansion of aquaculture ponds  through occupying cultivated lands in the countryside of China and analyze the influence of the process on local ESV and the function of cultivated lands.

It is really interesting issue, but it should be hypothesis formulated. After that the discussion will be connected with hypothesis, too.

The texts in figs,  should be write in bigger fonts.

You wrote about ecosystem services - in my opinion it should be add definision of this term and write more examples in part introduction.

Round 2

Reviewer 1 Report

The authors have appropriately considered my comments and satisfactorily addressed my concern regarding the conclusions presented in the previous submission. Therefore, I think the overall quality of the manuscript has "sufficiently improved to warrant publication in IJERPH".

This manuscript is a resubmission of an earlier submission. The following is a list of the peer review reports and author responses from that submission.